# Solar Energy Transformation Strategies by Ecosystems of the Boreal Zone (Thermodynamic Analysis Based on Remote Sensing Data)

**DOI:** 10.3390/e22101132

**Published:** 2020-10-06

**Authors:** Robert Sandlersky, Alexander Krenke

**Affiliations:** 1A.N. Severtsov Institute of Ecology and Evolution, Russian Academy of Sciences, Moscow 119071, Russia; 2Institute of Geography, Russian Academy of Sciences, Leninskiy Prospekt, 14, Moscow 119017, Russia; krenke-igras@yandex.ru

**Keywords:** energy balance, thermodynamic variables, exergy, vegetation work, information increment, succession, seasonal dynamic, evolution, order parameters

## Abstract

The hypothesis of an increase in free energy (exergy) by ecosystems during evolution is tested on direct measurements. As a measuring system of thermodynamic parameters (exergy, information, entropy), a series of measurements of reflected solar radiation in bands of Landsat multispectral imagery for 20 years is used. The thermodynamic parameters are compared for different types of ecosystems depending on the influx of solar radiation, weather conditions and the composition of communities. It is shown that maximization of free energy occurs only in a succession series (time scale of several hundred years), and on a short evolutionary time scale of several thousand years, various strategies of energy use are successfully implemented at the same time: forests always maximize exergy and, accordingly, transpiration, meadows—disequilibrium and biological productivity in summer, and swamps, due to a prompt response to changes in temperature and moisture, maintaining disequilibrium and productivity throughout the year. On the basis of the obtained regularities, we conclude that on an evolutionary time scale, the thermodynamic system changes in the direction of increasing biological productivity and saving moisture, which contradicts the hypothesis of maximizing free energy in the course of evolution.

## 1. Introduction

Beginning with Alfred Lotka [1,2], who was one of the first to try to determine the direction of evolution of living matter within the framework of thermodynamic theory, a clear idea of maximizing the energy used by the biosphere was formed as the goal of evolutionary development. According to the rule formulated by A. Lotka: “The direction of evolution is such that the total flow of energy passing through the system reaches the maximum value possible for this system” (cited from [3]). The “principle of maximum effect of external work” proposed by Ervin Bauer [4] is that the development of biological systems is the result of an increase in their external work—the impact of these systems on the environment. Based on this principle and the biogeochemical principles, Vladimir Vernadskiy [5], Vlail Kaznacheev [6] formulated the laws of Vernadskiy-Bauer: (1) geochemical biogenic energy tends in the biosphere to its maximum manifestation; (2) during the evolution of species, the organisms that increase their biogenic geochemical energy survive. In classical ecology, the law of Eugene and Howard Odum is better known—the law of energy maximization: “In competition with other systems, the one that best contributes to the flow of energy and uses the maximum amount in the most efficient way survives” [7]. The wording of this law by Nikolay Pechurkin [8] as “the energy principle of extensive development”, according to which, “in the processes of biological development of supraorganism systems (evolution, ecological successions and rearrangements), the amount of used biological energy flow increases, reaching local maximum values in stationary states.”

However, an increase in the flow of energy used by living matter leads to an inevitable increase in the production of entropy in the system, and, according to the second law of thermodynamics, to an inevitable thermodynamic equilibrium. This fact forced the leading physicists of the early twentieth century (among them were not only Alfred Lotka, but also Max Planck and Karl Heisenberg) to admit the importance of classical thermodynamics in describing living matter. Nevertheless, even then it was obvious that the disequilibrium of living matter is associated with the ability to extract ordering (negative entropy) from the environment [9]. It was then, due to the inapplicability of the law of increasing entropy in biology that the concept of fundamental impossibility of modeling the evolutionary process by physical systems arose. In the course of these discussions, the main ideas were formed how systems with living matter can function from the standpoint of physics—their key properties were recognized as openness and nonequilibrium.

The next stage in the development of the ideas about living matter was the appearance of the works of the Brussels school and the concept of dissipative and self-organizing structures. Ilya Prigogine in the mid-1950s formulated the principle of minimum entropy, according to which, in a system at a stationary state, internal nonequilibrium processes proceed in such a way that the increase in entropy is minimal. This means that the system, due to internal irreversible processes, is not able to leave the stationary state. However, this is true only under constant external conditions. When the external influence (flows entering and leaving the system) changes, the system leaves one stationary state and passes into another, if new external conditions persist over time. Accordingly, the criterion of the minimum production of entropy (dS/dt ≤ 0) of Prigogine–Glansdorf [10] was proposed as an evolution criterion for such systems. However, according to Ilya Prigogine’s co-author, Dilip Kondepudi [11], there is no general rule providing an “extremum principle” that governs the evolution of a system far from equilibrium to a steady state. Glansdorf and Prigogine argue that [10], irreversible processes are regulated by global extreme principles only in local areas, for which restrictions on their structure and macro parameters are defined. Thus, for non-stationary nonequilibrium processes, the principle is still the same, that is the second law of thermodynamics, according to which processes that are far from thermodynamic equilibrium, adapt to stable states in which they dissipate energy and produce entropy at the maximum possible level.

Despite the seeming simplicity of the statement that the principles of maximum and minimum entropy are not universal, it was not accepted by all researchers, which led to not very constructive discussions. Moreover, we got the impression that the division between the supporters of these principles is between bioenergetics working with cells and organisms [12,13,14], and ecologists who consider communities in their interaction with the environment (atmosphere) [15,16] but it’s not necessarily (for example [17]). For example, the entropy analysis of the basal metabolism of human body by Ishira Aoki [18,19] shows, firstly, a different rate of entropy production at different stages of life and a different direction of its production. In childhood, the body grows and maximizes the production of entropy, and after the transition phase, the rate of its production begins to decline. He obtained a similar course of the dynamics of entropy production for lake systems [20], on the basis of which he concluded that this pattern is universal [21]. In another time scale of the system, Keith Skene [22,23] analyzing the production of entropy in the succession series showed that the production of entropy reaches its maximum at its final stage—climax. However, he left out the post-climax, crumbling community.

In our opinion, the inconsistency of the results obtained by various researchers may occur due to the following factors: (1) wider homeostatic capabilities of organisms in comparison with ecosystems, (2) different degrees of openness of ecosystems and organisms, (3) hierarchical organization of living matter, at different levels and timescales various extreme principles are implemented. The “formal” reconciliation of extreme principles was carried out in a strictly thermostatic framework by Leonid Martyushev [24] and Roderick Dewar [25,26] who showed that the principle of minimum entropy is a special case of the principle of maximum entropy, which is realized at different stages and on different scales of the state of the system.

On the other hand, when comparing the results of various studies on real systems, we see an obvious problem of parametrizing the observed phenomena. The essence of this problem is that an object (organism or ecosystem) dictates to us its own way of description (macro parameters) available to our measurement. Accordingly, entropy as a parameter can be determined in numerous ways [27]:Phenomenological entropy is a component of heat exchange (model of the thermal machine).Statistical entropy is a measure of disorder during heat exchange.Entropy is a quantity of information that is transferred during communication processes (the theory of communications by Shannon / Ashby / Hartley).Entropy of a system characterizes it by the energy distribution of particles, i.e., it is a measure of connectedness and interaction inside or around a system.

The last of the given definitions is the most general and most informative from the point of view of the structural organization of the system. A very large body of papers has shown the relationship between different interpretations of entropy/information, for example [28,29,30]. On this basis, Alexander Hazen [31] gives a developed theory combining the principle of minimum entropy production in a stationary state and the principle of maximum entropy production as the basis for choice on an evolutionary time scale. Ultimately, at the present stage of development of physics and synergetic, in our opinion, there is no generalizing consistent physical theory of evolution. Moreover, if its existence is possible, then only with the use of an extended interpretation of entropy as the interaction of the system with the environment or with another system. The result of this interaction is the synthesis of information about its new state.

The desire to link information-disequilibrium with the amount used by living matter led to the development of ideas about the quality of energy in the ecosystem. In the second half of the 20th century, the concept of exergy entered thermodynamics. “Exergy is the maximum useful work that can be obtained when a working fluid or energy source comes into contact with the natural environment when reaching equilibrium with it” [32]. Exergy is determined by the degree of non-equilibrium of the system that converts energy, depending on its structure. The concept of exergy has been enthusiastically adopted in various fields of natural sciences from physiology to climatology. Sven Jorgensen [33,34,35,36], Yury Svirezhev [36,37] and James Kay with coauthors [38,39] can certainly be considered the pioneers of this direction in ecology.

After analyzing the literature on the use of the concept of exergy for living systems over the past decade, several main directions can be identified by different types of research objects. To begin with, all research can be divided into two large groups: exergy analysis of organisms [40] and eco-exergy analysis of communities and ecosystems. Several main areas can be highlighted in the last group: analysis of metabolism and nutrition [41,42,43], analysis of the efficiency of individual organs [44,45], analysis of exergy and lifespan [46,47], and human body comfort [48,49]. 

The application of exergic analysis in ecology can be divided into the following areas: exergy as indicator of ecosystem health and sustainable development [50,51,52,53,54,55,56], ecosystem sustainability and complexity [57,58] ecoeconomic [59], landscape-atmospheric interactions and climate-change assessment [60,61,62], ecosystem services assessment and landscape planning [63].

In accordance with the concepts of thermodynamics, all the energy entering the ecosystem is spent on useful work: on the creation of products, on the evaporation of moisture, on thermal dissipation (we can talk about maintaining the temperature of the environment), on maintaining and accumulating internal energy. The part of the incoming energy that is capable of doing useful work to maintain the system in a non-equilibrium state with low entropy is exergy. The “useful” work of the ecosystem is manifested in the maintenance and intensification of the water cycle in the biosphere and the provision of the bioproduction process. The rest of the absorbed energy is spent on increasing the internal energy of the system. In classical thermodynamics, internal energy is associated with the movement of molecules (heat exchange) and chemical bonds (internal energy). In an ecosystem, internal energy can be associated with the interactions of individuals of different species and parts of the system, with the maintenance of its internal structure, with the accumulation of energy within the system in partially closed exchange cycles. Apparently, the internal energy in the ecosystem can also be associated with soil-forming processes, in particular with the accumulation of carbon in the soil and maintaining its content at a certain equilibrium level. According to Sven Jorgensen and Yury Svirezhev [36], in the process of transformation, exergy is converted into energy that is incapable of doing useful work—bound energy—heat energy with high entropy and is removed from the ecosystem. Thus, the maintenance of organization (order) in the ecosystem is due to the dissipation of entropy into the environment in the process of energy conversion. The exergy of an ecosystem is a thermodynamic variable that reflects the relationship between the structure and its transformation of energy and, along with other thermodynamic variables, makes it possible to evaluate the peculiarities of the functioning of the system as a result of its structure. Assessment of exergy and heat flux of the active surface (temperature) gives a fairly complete picture of energy conversion, and the difference between absorbed energy and exergy reflects the change in internal energy. The exergy of the system is the higher the further it is from the region of the equilibrium state with a local maximum of entropy. This distance and, accordingly, the degree of its non-equilibrium can be estimated from the difference of the corresponding entropies and, more accurately, from the Kullback’s entropy [37], which reflects the increment of information or order in a non-equilibrium system in relation to an equilibrium one.

Within the framework of the modern thermodynamic approach, the functioning of a system with the participation of living matter according to James Kay and Roydon Fraser [39] is a transformation of exergy, and it is assumed that the development of the system is aimed at increasing the efficiency of its use, and evolution is the complication of living systems for more efficient use of exergy, that is moving away from thermodynamic equilibrium. According to them the criterion for evolutionary development is a decrease in heat flow from the active surface of the ecosystem that converts solar energy. In the course of competition, communities that most effectively cool the active surface of their canopy gain an advantage due to evapotranspiration (according to [64]). Thus, the lower the heat flux from the active surface and the higher the consumption of absorbed solar energy for evaporation, the higher is the evolutionary stage of the system. Sven Jorgenson and Yury Svirezhev [36], having carried out a deep analysis of applications of the ideas of thermodynamics in ecology and relying on extensive empirical material, developed the “preliminary fourth law of thermodynamics”, the essence of which is that maintaining the state of living matter and of related systems in a non-equilibrium stationary state is determined by the exergy flow. The Fourth Law of Thermodynamics is proposed to explain growth and development in ecological systems. In this case, growth is interpreted as an increase in the size of the system, and development—as an increase in the organization, regardless of the size of the system. 

Ultimately, the target function of a living substance is defined as an increase in exergy, that is, the ability to perform useful work. Even if we consider this as a hypothesis, its validation based on the study of real systems can be considered an important problematic area of ecology. A complete analysis of the food chain and species structure of the ecosystem, which allows such validation to be formally carried out, is, in the general case, an unsolvable task. Therefore, it is natural to compare the structure of energy flows and the transformation of exergy in an ecosystem with some well-observed and measurable functionally important elements of its own structure with direct or indirect consideration of its provision with moisture and elements of mineral nutrition. This opportunity is given to us by modern methods of remote sensing in various spectral ranges. Multispectral remote sensing data allow evaluating the operation of the main solar energy transformer—vegetation. The spectral structure of the reflected radiation in comparison with the structure of the incoming solar radiation per every elemental area (pixel, element of thermostatic system) allows estimating the thermodynamical parameters of ecosystems at the moment of measurements. The problem in the application of remote sensing data is the instantaneousness of measurements. Due to this, information is relevant only for its instant conditions (time of day, season, weather conditions). However, having a series of measurements of thermodynamic variables under different conditions, you can extract the “basic variables” (invariants), or order-parameters in terms of synergetics (by Herman Haken [65]). Order-parameters are determined by spatial-temporal variation of thermodynamic variables. Spatial-temporal variation of a specific thermodynamic variable is determined by the interaction of the order-parameters with “control parameters”: the influx of the solar radiation, weather, relief, condition of vegetation etc. Digital elevation model allows us to calculate the relief parameters for the various hierarchical levels, which determine the moisture and heat redistribution, and to explore it as a control parameter of the conversion of solar energy. Mass, composition and structure of vegetation, measured in the field measurements, allow us to investigate relationship between thermodynamic variables and structure of vegetation in ecosystem. The foregoing allows us to determine the purpose of the presented study: as a test of the hypothesis of maximizing the flow of free energy at different time scales for specific ecosystems of the boreal zone of the European Plain.

## 2. Materials and Methods 

The study was carried out on the territory of the Central Forest State Natural Biosphere Reserve (56°30′ N, 32°53′ E) located in the southwestern part of the Valdai Upland, within the Main watershed of the European (Russian) Plain (Figure 1a). The natural complex of the reserve is typical for the southern taiga subzone and is the standard of the primary vegetation cover of the vast area of moraine relief in the central part of the Russian Plain. The territory is also characterized by extensive raised bogs, and in the protected zone of the Reserve, communities are represented by various stages of successions in the place of clearings of different ages and meadow communities in the place of abandoned arable lands and pastures (Figure 1b). The territory is provided with ground-based measurements of the state of vegetation and soils obtained as a result of scientific research carried out by various teams of scientists for more than fifty years, including an extensive network of field descriptions of ecosystems, meteorological data, digital elevation model, etc. The uniqueness of the territory and its provision with field data and long-term observations allow one to study the spatio-temporal variation of thermodynamic characteristics depending on the state of the natural vegetation cover and its constituent communities, which is determined by the seasonal course of incoming solar radiation, weather conditions, and vegetation properties. The presence of a digital elevation model makes it possible to assess its impact on the transformation of vegetation by communities and assess the contribution of vegetation composition regardless of the contribution of the relief.

In this work Landsat TM and ETM multispectral imagery was used as the measurement system. We used 20 separate cloudless scenes, performed in the morning in different seasons from 1986 to 2009 (Table 1). The survey is made in seven spectral ranges: six channels in short-wave (spatial resolution of 30 × 30 m per pixel) and long-wave thermal field (resolution of 60 × 60 m). Landsat survey channels correspond to the main windows of atmospheric transparency, which allows virtually excluding its impact, covering 22.8% of the length of the solar spectrum and describing 42.4% of the emission intensity of the solar constant of the spectrum. Landsat TM and ETM + spectral channels are: 1—blue (0.45–0.515 µm), sensitive to the atmosphere transparency, the energy in this range is absorbed by chlorophyll; 2—green (0.525–0.605 µm), the peak of reflectivity of the leaves, 3—red (0.63–0.69 µm), absorption by chlorophyll b, 4—near infrared (0.77–0.90 µm), reflected by mesophyll, 5—medium infrared (1.55–1.75 µm), sensitive to water and lipids in leaf tissues, 7—far infrared (2.09–2.33 µm), which is absorbed by vegetation and soil moisture, 6—long-wave thermal (10.12–14.5 µm) range, in which the temperature of the active surface is measured. According to calibration constants of the sensors [66], the brightness values in the channels are converted into radiation, reflected by the active surface (W/m^2^). The brightness value of the thermal channels is converted into heat flux from the active surface and its temperature. Incoming solar radiation in the bands is calculated according to the solar constant for each band adjusted to the solar angle and the distance between the Sun and the Earth at the time of the survey. Accordingly, the energy absorbed in each band is calculated as the difference between the incoming and outcoming solar radiation.

Thermodynamic properties of ecosystems are calculated according to the methodology offered by Sven Jorgensen and Yury Svirizhev [36] and corrected for the Landsat satellites [67]. In its plainest form, energy balance of non-equilibrium system (B) includes exergy (Ex), bound energy—which is incapable of conversion to useful work (STW) and internal system’s energy increment (U):B = Ex + STW + U.(1)

As to the simultaneous assessment of thermodynamic variables during the single scene shooting, it is more correct to speak of the increment of these values rather than absolute meanings. Exergy is the energy that can be converted into useful work. As regarding ecosystems, exergy closely relates to sustaining of water cycle. Bound energy is the energy that is dispersed into the environment together with a heat flux and entropy. Internal energy increment means energy accumulation, probably in the form of organic matter upbuilding. Exergy is evaluated as a function of non-equilibrium of incoming and reflected solar radiation specters (increment information by Kullback). The more the specters converge, the more equilibrium the ecosystem receptor is to the flux of incoming energy, hence the information increment is smaller. The information increment for the Landsat satellite imagery (K, nit) is calculated as:K = ∑p*_i_^out^*lnp*_i_^out^*/p*_i_^in^*,(2)
where p*_i_^out^* = e*_i_^in^*/E*_in_* is the ratio of incoming energy (e*_i_^in^*) in the spectral band (*i*) and the total incoming energy is (E*_in_*); p*_i_^out^* = e*_i_^out^*/E*_out_* is the ratio of reflected energy (e*_i_^out^*) in the spectral band (*i*) and total reflected energy (E*_out_*).

The exergy of solar radiation (Ex) is calculated as:Ex = E*_out_*(K + lnA) + B,(3)
where E_in_ is incoming solar radiation, W/m^2^; E_out_ − reflected solar radiation, W/m^2^; B = E_in_ − E_out_ is absorbed energy; and A = E_out_/E_in_ is albedo.

To evaluate bound energy (energy dissipation with a heat flux and entropy), it is necessary to assess the entropy of reflected solar radiation. The larger is the entropy of the reflected solar radiation, the more equilibrium its flux is. The entropy (Sout, nit) is calculated as:S = − ∑p*_i_^out^*lnp*_i_^out^*,(4)Bound energy (STW, W/m^2^nit) is evaluated as:STW = TW × S,(5)where TW is the heat flux from the active surface, captured by a heat channel.

The increment of system’s internal energy (DU) is a transition of absorbed solar energy into internal system’s energy. It is evaluated as a residue from the “balance” equation of absorbed energy (B):DU = B – Ex – STW,(6)To assess the energy consumption for biological productivity, we used a standard channels’ ratio – the difference between reflected energy in the red (RED) and short-range infrared ranges (NIR):VI = NIR – RED.(7)

Thus, the following thermodynamical characteristics were calculated: the forming of the balance of the absorbed solar energy (W/m^2^), which is exergy (W/m^2^), bound energy (W/m^2^nit) and internal energy increment (W/m^2^); structural characteristics of system, describing its nonequilibrium state, which are Kullback information increment (nit) and entropy of outcoming solar radiation (nit); system heat flux (temperature) and vegetation index (W/m^2^).

To estimate the contribution of the relief into energy transformation we used the morphometric characteristics calculated for different hierarchical levels on digital elevation model with the resolution of 30 × 30 meters per pixel. DEM is based on 1:10000 scale topographic maps. Hierarchical levels of relief were calculated with the help of spectral analysis based on two-dimensional Fourier transformation [67,68], which allowed us to represent the absolute relief heights in the shape of a set of waves, decompose them according to separate frequencies and characterize the degree of manifestation of each frequency by its amplitude. By using the inverse Fourier transformation, we singled out and calculated the surface for the following hierarchical levels: mesorelief of different orders, with linear dimensions/amplitude of heights of 3810/80 m, 1050/50 m, 450/30 m, 270/15 m, and micro-relief 150/5 m. For absolute elevation of each hierarchical level we calculated morphometric characteristics which determine the redistribution of solar radiation, heat and moisture: slope (the first derivative, maximum slope of the surface); laplacian (second derivative, surface shape—convexity/concavity); shaded relief from the east and from the south at the solar altitude of 45°; convexity (profile, plan, cross sectional); curvature (minimum and maximum).

To analyze the dependence of the thermodynamic characteristics on the weather conditions, we used data from the weather station of the Central Forest Reserve: air temperature on the survey day and the total precipitation that occurred before the remote sensing survey, accumulated temperatures and precipitation totals for the periods of 36, 24, 12, 6 and 3 days before the survey.

The work used the materials collected on the territory of the Central Forest Reserve from 1998 to 2011 by the IPEE RAS team, graduate students and students of the Geographical faculty of MSU. Complex descriptions of vegetation were carried out according to the standard methods and referenced on the ground using GPS. The following characteristics were used directly in the analysis (1300 descriptions): forest stand height (average and by canopy level, m), absolute density (proportion of 1), timber stock (for each species, m^3^/ha), projective cover (%) for layers (moss and herb). 

The materials used (thermodynamic characteristics, morphometric characteristics, field descriptions) were combined into a single geographic information system using geospatial techniques. The analysis of the dependence of thermodynamic characteristics, their spatial variation and degree of organization on the weather conditions were obtained from the meteorological station of the reserve and the influx of solar radiation was carried out by the methods of multiple regression and nonlinear estimation. The thermodynamic variable of each period was used to calculate the distribution parameters for the study area as a whole and for generalized types of ecosystems—forests, meadows and raised bogs, and the dependence of their seasonal variation on the incoming radiation (solar constant), phonologic phase (day number from the beginning of the year (DOY)) and meteorological variables.

To highlight the parameters of the order of space-time variation, the method of principal components (factor analysis) was used. Herman Haken showed [65] that factor analysis is a reliable way to isolate them. The multidimensional space is formed by thermodynamic variables: absorbed energy, exergy of reflected solar radiation, heat flux, entropy of reflected solar radiation, increment of information, vegetation index for twenty periods were transformed using principal components analyses into the space of independent order parameters.

The assessment of the contribution of morphometric characteristics to the order parameters was carried out by the method of stepwise multiple regression. In addition to the coefficient of determination (R^2^) of the parameter by relief and the sign of influence for a significant morphometric characteristic, for each parameter their values predicted from the relief and regression residuals were obtained – that part of the variation of the parameter, which is mainly determined by the intrinsic state of the vegetation. The analysis of the contribution of vegetation properties was carried out differentially for each parameter. Then, for each order parameter, its relationship with the characteristics of vegetation was estimated.

## 3. Results

Comparison of the seasonal variation of the thermodynamic characteristics, calculated according to Landsat TM and ETM + multispectral survey for the territory of the Central Forest Reserve for various years with the seasonal variation of the thermodynamic characteristics, calculated according to MODIS multispectral survey for 2002, cell 0.5 × 0.5° by our team [27] showed their high similarity. The similarity in seasonal variations subject to the differences in the set of measured spectral bands indicated the applicability of the proposed approach in a wide range of scales. The analysis of the dynamics in the thermodynamic variables showed that for the biosphere as a thermodynamic system, there are two relatively independent subsystems: subsystem that is responsible for the absorption of incoming solar energy, exergy and exergy conversion into heat flux and informational subsystem that is defined by entropy, information increment and biological productivity.

### 3.1. Seasonal Dynamic and Weather Influence 

Analysis of the seasonal variation of energy variables for the area as a whole showed that the energy conversion is determined by incoming solar radiation and, therefore, is fundamentally different for the snow and vegetation period. In winter, the system converting the energy is at most close to equilibrium. In the snowless period with an increase in income of solar radiation, expenditure of energy on exergy maximize, entropy decreases, and the increment of information and biological production related to the increase of non-equilibrium. In general, for the landscape the solar energy absorption, exergy and non-equilibrium are maximal in summer (June). The analysis of the dependence of seasonal variation of the thermodynamic parameters on the whole territory under observed weather conditions (Figure 2) showed that the absorption and exergy (Figure 2a) are almost independent from weather conditions and are mainly determined by income of solar energy. The temperature (heat flux) is determined by incoming solar radiation and temperature of air mass (Figure 2b). Non-equilibrium of vegetation cover (Figure 2c) weakly depends on incoming solar energy, weather conditions and the seasonal state of the vegetation cover (phenological phase). On this background, the effect of weather conditions is visualized: simultaneous growth of humidification and warming, with the phenological phases, puts the system, which converts the solar energy, into the most non-equilibrium state. Biological productivity depends both on the solar radiation income with the weather conditions and on the phenological phase (Figure 2d). Meanwhile spring variation of the biological productivity is poorly described by external variables, which apparently indicates that the state of the landscape responds to the increase in the income of solar radiation with delay.

An assessment of differences in the spatial variation on the root-mean-square deviation for each thermodynamic variable of each survey helped identify variables with different scales of spatial variation. The smallest spatial variation is held by absorbed solar energy and the components of its balance, and the largest one is the characteristic of the variables describing non-equilibrium of energy conversion and biological productivity. Comparison of the seasonal dynamics of the spatial variation of the variables shows that the absorption of solar radiation, exergy and the increment of the internal energy maximize their spatial variation in early spring, during the snow cover melting, and minimize it during the vegetation period, but the information increment of the outcoming radiation, bound energy and biological productivity, on the contrary, increase their spatial variation during the vegetation period. Under steady moistening, absorption of solar energy and exergy reduce their spatial variation. The spatial variation in temperature decreases with large amount of precipitation and increases with growth of accumulated heat. The information increment maximizes its spatial variation with an increase in income of solar radiation and with the increase in the information increment, and the spatial variation of entropy, however, is largely associated with the weather conditions, particularly precipitation. Stress because of excessive amount of precipitation, makes it necessary for vegetation to adapt to various local conditions. This adaptation is carried out, among other things, due to structural changes in the system, reflecting in the change of entropy, and the information increment. The spatial variation in biological productivity, as well as the information increment increases with the growth of solar radiation income and vegetation index. Against this background the positive effect on the spatial variation of biological productivity causes the growth of moistening.

Let’s consider the variation of variables for the main types of ecosystems: forests, swamps and meadows and its dependence on the weather. In winter period meadow communities and bogs unlike forests are in a state close to thermodynamic equilibrium. As we can see from Table 2 only forest communities maximize the absorption of solar radiation and exergy and minimize heat flux (temperature) and the energy dissipation. Meadows maximize biological production, internal energy and the information increment for the entire snowless period. Bogs maximize heat flux and energy dissipation and minimize the information increment while supporting the biological production at average level. The temperature over the bog during the entire snowless period is by the average of 3.4° higher than in the forest and by 1.8° higher than in the meadows. In summer the bog is warmer than the forest by the average of 4.5° and the meadows by 2.2°. 

Thus, each type implements its strategy of solar energy converting: the meadow and the bog minimize losses by evaporation; meanwhile the bog is the most equilibrium throughout the snowless period at the same time ensuring a sustainable production. It is characteristic that the meadows and the bogs accumulate significantly more internal energy in comparison to the forest. Coefficient of variation as a measure of the spatial variation of the thermodynamic variable allows allocating variables with minimal spatial variation for each type of community, that is, the variables that are the least sensitive to variations in habitat conditions and the composition of the vegetation itself. It is characteristic that no forest is distinguished by a variable with maximum and minimum coefficient of variation, despite the fact that these communities occupy the greatest area. For meadows maximum variation throughout the whole snowless period is noted for the absorption of radiation and temperature. The bog vegetation holds the best regulatory abilities and supports a minimum variation of almost all variables.

An analysis for sensitivity of the thermodynamic variables for each type of community has shown that the energy absorption and exergy by forest depend only on the income of solar radiation and are insensitive to changes in the weather (Figure 3a). The radiation absorption in the meadows is positively influenced by accumulated precipitations, while in the bogs their effect is negative. Exergy both in the meadows and the bogs decreases with the growth of temperature accumulated that points directly at the economy of moisture and under the excess of heat supply, they evaporate significantly less moisture than forests. Heat flux/temperature (Figure 3b) for different communities is determined by the prevailing air mass and accumulated precipitations, at the same time big amount of precipitation that fell in the meadows, almost immediately reduces heat flux, while the rest of the communities do not react to them. During the snowless period, for the forests and meadows the information increment (Figure 3c) is negatively correlated with the income of solar radiation, and for the bogs information increment does not depend on it. For the meadows and forests, the amounts of accumulated temperatures and precipitations increase the information increment, but for the bogs, on the contrary, they reduce it, transferring them into a more equilibrium state. The dependence of biological production on the weather conditions (Figure 3d) is the same for all types: on the background of the positive impact of incoming energy it is increased by the growth of heat and moisture provision. 

To assess the organization of the system (consistency of changes in variables), one can use the determinant of the correlation matrix, calculated within the framework of the principal component method for thermodynamic variables of each term. The logarithm of the determinant (Δ) multiplied by minus one in the sense is identical to the information in the system:I = − logΔ.(8)

The measure of organization of the system for the landscape as a whole is most positively associated with the information increment, index of biological productivity and temperature which allows to define the objective function of the system as the maximization of non-equilibrium of conversion of solar energy, biological production and the heat flux from the active surface to the atmosphere. Forests have the maximum of organization, and they are characterized by the maximum dependence of organization measure on the variation of the same thermodynamic variables as for the landscape on the whole. The organization measure of meadows is minimal, but the correlations are similar to those for the forest. Unlike the other two types the productivity index for bogs is weakly associated with the organization measure and other correlations are much less than for forests and meadows. Evaluation of the dependence of organization measure on the solar radiation income, the day from the beginning of the year, which characterizes the phenological phase, and on the weather conditions show that the organization of the thermodynamic system is largely determined by phenology that is the calendar phases of vegetation self-development. Comparison of these estimated measures among the types of communities showed that the functioning of life form of mosses and herbs is less than of the trees determined by phenology. Estimation of whether organization is connected to weather conditions or not, shows the tendency of its increase with the increase of temperature and precipitations. Thus, in the considered set of properties we can assume that the forests are thermodynamically most open and due to high heat losses by evaporation and creation of the actual climate are not very sensitive to changes in weather conditions over time. Bogs, on the contrary, can be considered as the most closed systems that hold the thermodynamic similarity of function over the entire area and provide economical moisture consumption and sustainable production throughout the vegetation season. Meadows hold intermediate position in accordance with isolation and they maximize biological production along with economical expenditure of moisture.

### 3.2. Order Parameters of Thermodynamic System

Spatial-temporal variation of the thermodynamic variables under study is defined by three order parameters that describe 65% of the variation of variables. The first parameter describes 43.5% of the variation, the second and the third describe 15.2% and 5.9%, respectively. The first parameter (Figure 4) has a positive effect on the solar energy absorption and exergy throughout the year, a negative effect on heat flux and, to a lesser extent, on the biological production. The second and third parameters describe the information increment and biological productivity with a negative sign and the entropy with a positive sign: the second, in the summer (Figure 5); the third, in spring and autumn (Figure 6). Analysis of the dynamics of contribution of each order parameter to the variation of each thermodynamic variable shows that in summer, when the thermodynamic system is the most non-equilibrium and most organized, the variables are defined by parameters more clearly.

Evaluation of order parameters for the main types of vegetation communities (Figure 7a) shows that a prevalent order parameter exists for each type: forests are, on average, characterized by a high value of the first parameter, the third is characteristic of meadows, and bogs are associated with the second. Placement of types in the coordinates of parameters (Figure 7b) and the distance between them demonstrate that the types of communities form completely distinct subsets, the most isolated among which are bogs. The strict association between parameters and vegetation types in generic form determine the control parameter as a possible shift in space and time of forests, meadows, and bogs. In this case, the relatively high affinity of forests and meadows directly points at the existence of transitions between them in time: windfalls, burns-out, and felling are gradually replacing forest parameter correspondingly. Waterlogging as a process that occurs on a prolonged environmental time scale is also a natural control parameter, and the demonstrated closure of the bogs and fundamental differences in the functioning of forest and meadows determine discreetness of their boundaries.

Although an analysis of each parameter’s dependence on the morphometric characteristics of the relief was performed, in this work these results are clearly redundant. The first parameter is described by the relief by 20%; the second, by 27%; and the third, by only 5%. To assess the intrinsic contribution of the vegetation cover, we used the order parameter values that are free from the influence of the relief (the regression residuals by morphometric variables of the order parameters).

### 3.3. Vegetation Condition as Control Parameter of Thermodynamic System 

The calculated dependence of the order parameters on the properties of vegetation allowed to estimate the contribution of the structure of plant communities to transformation of the energy. Table 3 shows Spearman’s correlations between order parameters and vegetation cover, and plots for main vegetation properties are shown in Figure 8. Throughout the year, the forest increases absorption of solar energy and exergy, reducing non-equilibrium and biological productivity in summer and increasing them in spring. Dense forest cover reduces absorption and exergy and increases productivity. Young forests in general maximize biological productivity, and the middle-aged forests maximize exergy; however, old forests reduce both. Absorption and exergy are increased by the growth of coniferous portion of the forest stand composition, and non-equilibrium and productivity are increased by the proportion of deciduous trees (Figure 8a,b). The degree of development of the herb layer (Figure 8c) affects the energy conversion in full compliance with sub-area occupied by meadow communities in order parameter space. The more developed the herb layer is, the greater the non-equilibrium and productivity are, and lesser the absorption and exergy. The effect of the degree of development of moss layer (Figure 8d) on the conversion of energy is inverse to the influence of the grass abundance. 

Analysis of dependence of the order parameters on the species composition of the forest allowed not only to validate the indicated dependences, but also to reveal their mechanisms: the influence of broad-leaved species (mainly elm and linden) on conversion of solar energy is due to, first of all, the position occupied by them in relief. The thermodynamic system formed by deciduous species considerably differs from the system formed by conifers (mainly spruce): leafy species, under all other conditions being equal, obtain less exergy and more information increment and biological productivity.

Dichotomic classification of the territory according to order parameters allowed to identify eight classes of communities with different conversion of solar energy (Figure 9). Analysis of the seasonal dynamics of thermodynamic variables for the identified classes of the system allowed evaluating their variation in the successional series: meadows—destroyed forests—deciduous forests—coniferous forests. In general, during the vegetation period, non-equilibrium and productivity decrease from meadows to coniferous forests, but absorption and exergy increase. Within the forest communities, the differences in the heat flux in successional series are observed only in spring and autumn, when the deciduous forests may be warmer than coniferous by 1 degree; however, in summer, with significant differences in exergy, temperature in coniferous and deciduous forests differ only slightly. Thus, deciduous forests with higher productivity expend less energy to maintain circulation of moisture than coniferous forests.

## 4. Discussion

Our data on the variation of thermodynamic characteristics in space and time generally correspond to the measurement results obtained by other methods, including the data from FLUXNET Eddy covariance [69,70]. In the latter research [71,72,73,74], it is shown that as the age of forest communities increases, the production of entropy increases as the climax is approached, and then, as the community is destroyed, it decreases. This pattern was obtained for lake communities by Ichiro Aoki [20] and proposed by him as a universal law of the development of living systems [21]. The generality of this law is also confirmed in the environmental works by Keith Skene [22,23] and Roderick Dewar [25,26]. The variation of the entropy of the reflected solar radiation, which is directly proportional to the increment of information, fits well into this scheme, reaching a maximum in mature forests and decreasing towards meadows. The explanation for this lies in the increase in albedo with the age of the forest and, accordingly, with a decrease in temperature and an increase in heat flow. Comparative analysis of the variation of entropy in the raised bog and in the blueberry spruce forest according to long-term observations of Eddy covariance [75] generally confirms the relative independence of the bog functions from the influx of radiation. Bogs, being non-equilibrium ecosystems, have been shown to demonstrate unique thermodynamic behavior, which is fluctuant and strongly dependent on the moisture supply.

Analysis of temporal dynamics and spatial-temporal variation of thermodynamic variables showed that two relatively independent main subsystems are distinguished in the thermodynamic system of the southern taiga landscape: the subsystem responsible for the absorption of incoming solar energy, energy consumption for evapotranspiration and its transformation into heat flux and the “spectral” subsystem, which determines the spectral structure of the surface absorbing solar energy, and the increment of information and biological productivity closely related to this structure. We also demonstrated the presence of these subsystems for another measuring system, MODIS, for the regional [61] and global [60] scale. The operation of the first subsystem responsible for evapotranspiration, as shown by the results of the analysis of dependency of thermodynamic characteristics on the weather, is mainly determined by the input of solar radiation, which fully corresponds to the model results obtained by Axel Kleidon on the extensive empirical material [76,77]. The complex dependence of the operation of the production subsystem assessed by remote sensing data on external conditions is shown, for example, in [71,78]. The existence of two relatively independent subsystems with different functional significance in a thermodynamic system does not allow us to consider the exergy for vegetation as a certain integral indicator of useful work. Exergy reflects only “useful work” in transporting water from the soil to the atmosphere. The second type of “useful work” is biological productivity; due to the relatively low energy consumption, it is not significantly reflected in exergy and is actually described by an increment of information. It is typical that the adaptation of plants to environmental conditions, ensuring their stability, is achieved, in fact, in three different ways: maximization of exergy, maximization of biological productivity, and reduction of fluctuations in biological productivity over time with an actual lengthening of the growing season.

Thus, maximization of exergy [36] and, accordingly, minimization of the heat flow [39], cannot be considered as the general goal of the evolution of an ecological system in various ecological and evolutionary time scales: seasonal dynamics, successional dynamics, phylocenotic dynamics [79], short ecological time, long ecological time, and evolutionary time [80]. Under specific climatic conditions and seasonal dynamics, different types of communities implement different strategies. Maximization of exergy is realized in the succession series “grasses—small-leaved forests—spruce forests”. In a short evolutionary time of several thousand years, in the conditions of the southern taiga, the strategy of maximizing exergy and the strategy of “raised bogs” are realized in parallel, which is manifested in their progressive occupation of the territory.

Considering the tendency in changes in the thermodynamic system on an evolutionary time scale in a first approximation, one can determine the time sequence of the emergence and distribution of various life forms of plants. Thus, communities formed by conifers arose and became dominant in the Permian period, 300–250 million years ago [81]; deciduous (angiosperms) appeared more than 200 million years ago and occupied a dominant position in vegetation less than 150 million years ago (Cretaceous period). The age of the boreal forests is approximately 30 million years (end of the Paleogene). Grasses appeared 60 million years ago (the beginning of the Paleogene) and began to form large communities no earlier than 10 million years ago (Neogene), which is due to the aridization of the climate [82]. Despite the fact that mosses essentially did not changed since their emergence on land about 450 million years ago (Ordovician), as communities, raised bogs formed by them are relatively young. Sphagnum mosses became a coenosis-forming taxon only in the Neogene, 5 million years ago [81]. Thus, it can be seen that, if we do not consider sphagnum bogs, evolution on a geological time scale is definitely aimed at a more economical use of moisture, which is confirmed by studies of the evolution of the conductive tissue of vascular plants [83], aimed at increasing the “hydraulic efficiency” of moisture transport. It can be assumed that the development of raised bogs was a response to the increased manifestation of seasonality in fluctuations in precipitation and temperatures in the climate of middle latitudes, which occurred approximately 15 million years ago (the end of the Paleogene). The ability of sphagnum mosses to vegetate during the entire snowless period gave them obvious advantages over trees and grasses, while raised sphagnum bogs as a community is able to “save moisture” more efficiently than grassy communities. Thus, we can assume that on an evolutionary time scale, the thermodynamic system changes in the direction of increasing biological productivity and saving moisture. All this directly contradicts the hypothesis of exergy maximization as formulated by Sven Jorgensen, Yury Svirezhev [36] and Jamas Kay and Roydon Fraser [39]. However, an increase in biological productivity is an obvious increase in free energy (exergy) for the implementation of the main function of living matter: its self-reproduction. Apparently, this process is associated with an increase in the efficiency of moisture usage and an increase in internal energy, which is possibly associated with the accumulation of humus and peat (swamps), which increase the moisture capacity of the environment.

The evolution of the thermodynamic system of the vegetation cover appears to be prompted by the general trend of climate change. As a result of this process, diversity and the stability of biological productivity increase in a wide range of current climatic conditions. The combination in space of thermodynamic systems with different scales of moisture and heat fluxes into the surface layer of the atmosphere should significantly increase its turbulence and, accordingly, improve gas exchange. On the other hand, a very large difference in the space of the heat and moisture flux between the forest and raised bogs should increase the turbulence of a significant height of the surface layer of the atmosphere, increasing the efficiency of “biotic pump” by Victor Gorshkov and Anastasiya Makarieva [84]. Thus, the evolution of the thermodynamic system of the vegetation cover possibly generates synergistic effects that increase the useful work of the large “landscape—atmosphere” system as a whole, both in terms of increasing biological production and intensifying the moisture cycle. The latter, apparently, is associated with an increase in exergy, but at a higher level of organization of the thermodynamic system, which is not considered in the present work.

Analysis of temperatures and precipitation at the Reserve meteorological station since 1963 shows a well-pronounced trend of climate warming over the past 45 years [85]. Warming occurs primarily in winter and spring: according to the long-term norm, stable snow cover is established in the second half of November, and in recent years, in the second half of December; at the same time, earlier spring snowmelt has been observed in recent years. At the same time, the amount of precipitation is decreasing in spring and increasing in other periods. It can be assumed that, in terms of its parameters, the climate of the territory shifts towards the Atlantic period (6–7 thousand years ago), which, according to dating, marks the beginning of the formation of raised bogs on the territory of the Reserve [86]. Our analysis of the dynamics of the Reserve’s vegetation cover in recent years [87] and field observations show an increase in the renewal of broad-leaved tree species (maple, linden, hazel) in well-drained and warm relief positions and the expansion of sphagnum mosses in well-lit glades and under the forest canopy on flat and concave landforms. We also recorded an unambiguous increase in the areas of raised bogs, accompanied by waterlogging and drying out of adjacent forests. Thus, apparently, climate warming, which is mainly expressed by the lengthening of the growing season and from which sphagnum communities benefit, leads to an increase in the area of bogs and, accordingly, an increase in the heat flow. Accordingly, a positive feedback effect arises between the increase in precipitation and temperatures and the growth of raised bogs. At the same time, with a high amount of precipitation, and, accordingly, the level of the water, the swamps evaporate almost like an open water surface, which partly compensates for their warming effect. Climate softening stimulates the nemoralization of the forest vegetation of the Reserve, contributing to an increase in the average heat flow for the territory and a decrease in evaporation. In recent years, apparently due to the lack of spring precipitation, focal drying of spruce has been observed, which also indicates the degradation of climax communities under climate change. Climax communities strive as much as possible to ensure the maintenance of thermodynamic equilibrium between the soil and the atmosphere through the maximum possible evaporation. Apparently, when approaching thermodynamic equilibrium, a decrease in internal energy leads to a loss of stability and the ecosystem collapses in one way or another, and goes into a repeated cycle of self-development or, when the external control parameters (in our case, the climate) change, switches to another development trajectory.

## 5. Conclusions

Comparison of the dynamics of energy conversion for communities formed by plants with various life forms and their “target functions” showed that the forest communities is tend to maximize the functioning of the thermodynamic subsystem responsible for the absorption of solar energy and energy consumption for evapotranspiration; and the functioning of meadow communities and raised bogs, to a greater extent, at the stability and efficiency of the subsystem responsible for biological productivity. Forest communities act as the most open of the studied ones, a system that maximizes the absorption and consumption of energy for evapotranspiration in proportion to the increase in energy entering the system. The “openness” of coniferous forests is demonstrated to be noticeably higher than deciduous ones. Meadow communities maximize biological productivity with relatively economical use of moisture. The raised bogs maintain high disequilibrium and biological productivity by a quickly responding to changes in environmental conditions. Unlike meadow communities, the state of which depends on the phenological phase, bogs have a labile structure of spectral absorption that is capable of maintaining disequilibrium at approximately the same level with the change in weather conditions and solar radiation input. In spring, the swamp begins to work intensively immediately after the snow cover melts; in the fall, it continues to work, and in summer, with excessive heat supply, it limits its activity, reducing evaporation and photosynthesis, increasing energy dissipation into the environment and its accumulation. This lability provides a quick adaptation of the thermodynamic system of bogs to changing conditions (including weather) and sustainable production of biological matter at a sufficiently high level, as well as the accumulation of internal energy, including that in the form of dead organic matter.

Unfortunately, the increase in exergy in the process of self-development of complex systems, proposed as the fourth thermodynamic law, is not a universal invariant. In reality, it is much more complicated. At the same time, it is obvious that an extended thermodynamic analysis of the functioning of the plant cover based on multispectral information is very promising for a deeper understanding of the ecosystem processes. Comparative analysis of the functioning of various types of ecosystems formed by plants that appeared at different stages of evolution can provide important information for understanding the general direction of the evolutionary process.

In conclusion, it is necessary to pay attention to the huge role of forest vegetation in the regulation of heat flow and associated energy of the biosphere. Any deforestation increases the temperature by almost four degrees Celsius. This scale of influence on the climate is obviously quite comparable to any other factor. It is important to note that the development of raised bogs in the boreal zone over the past six thousand years also inevitably contributed to the heating of the atmosphere.

## Figures and Tables

**Figure 1 entropy-22-01132-f001:**
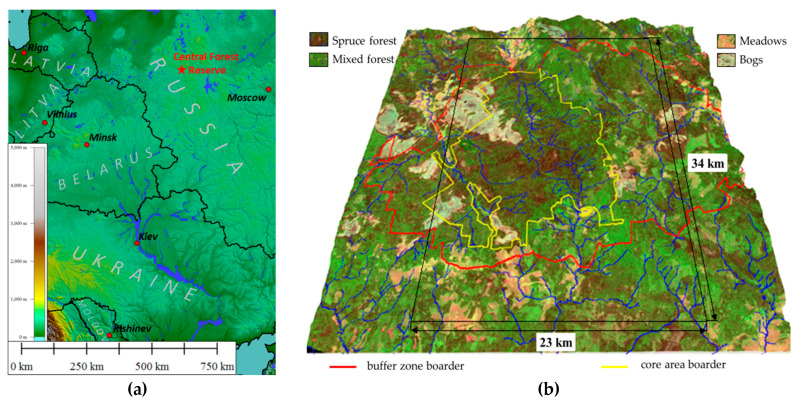
Study area: (**a**) Central Forest Reserve geographical position; (**b**) Central forest reserve, Rapid Eye image 02.09.2009 (false color, spatial resolution 6.5x6.5m) on digital elevation model.

**Figure 2 entropy-22-01132-f002:**
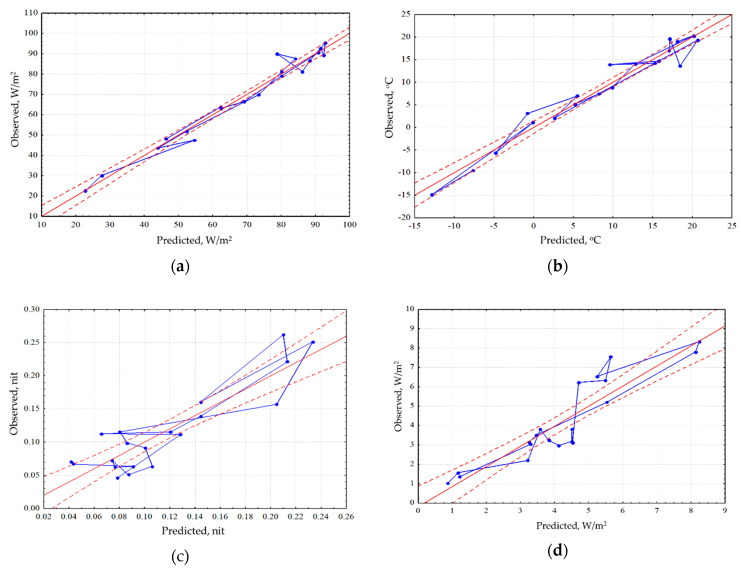
Dependence of seasonal variation of the thermodynamic parameters on the whole territory under observed weather conditions (regression for means of meteorological variables): (**a**) Exergy of solar radiation; (**b**) Temperature; (**c**) Information increment; (**d**) Vegetation index.

**Figure 3 entropy-22-01132-f003:**
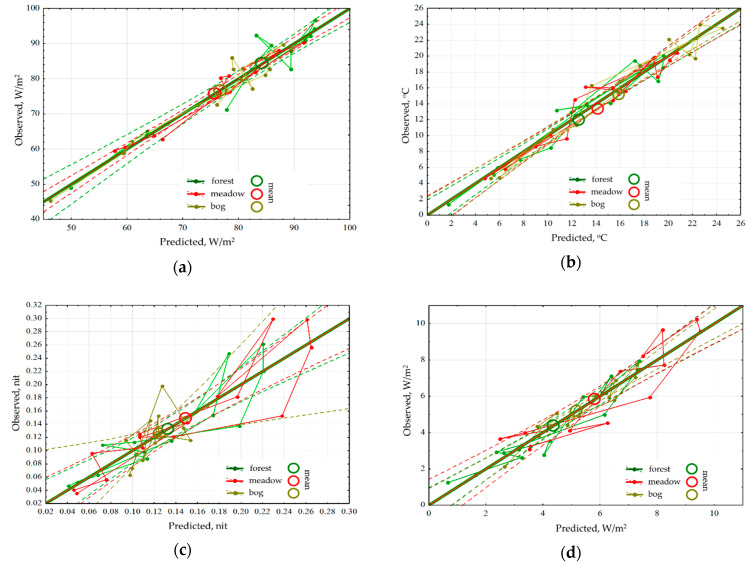
Dependence of seasonal variation of the thermodynamic parameters for the three main community types under observed weather conditions (regression from meteorological variables): (**a**) Exergy; (**b**) Temperature; (**c**) Information increment; (**d**) Vegetation index.

**Figure 4 entropy-22-01132-f004:**
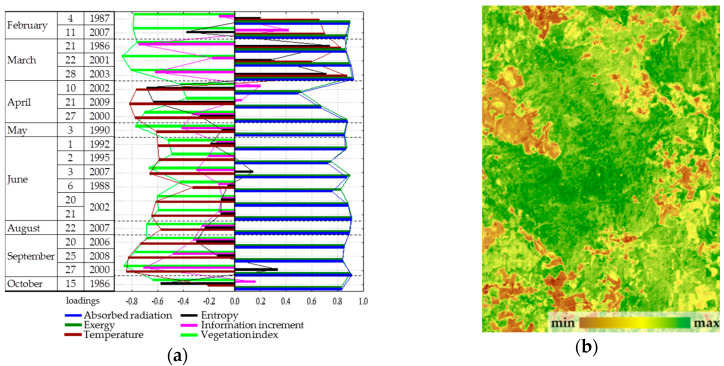
First order parameter—absorbed solar radiation and exergy throughout the year: (**a**) Factor loadings; (**b**) Spatial distribution.

**Figure 5 entropy-22-01132-f005:**
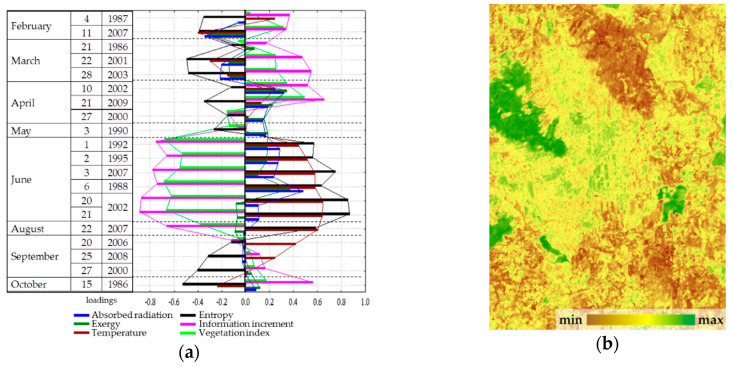
Second order parameter—non-equilibrium and biological productivity in summer: (**a**) Factor loadings; (**b**) Spatial distribution.

**Figure 6 entropy-22-01132-f006:**
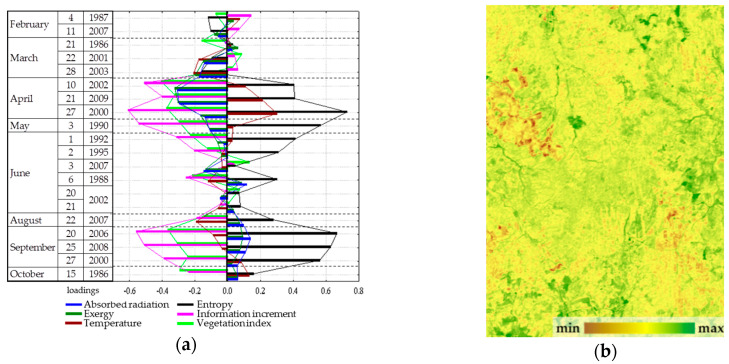
Third order parameter—non-equilibrium and biological productivity in spring and autumn: (**a**) Factor loadings, (**b**) Spatial distribution.

**Figure 7 entropy-22-01132-f007:**
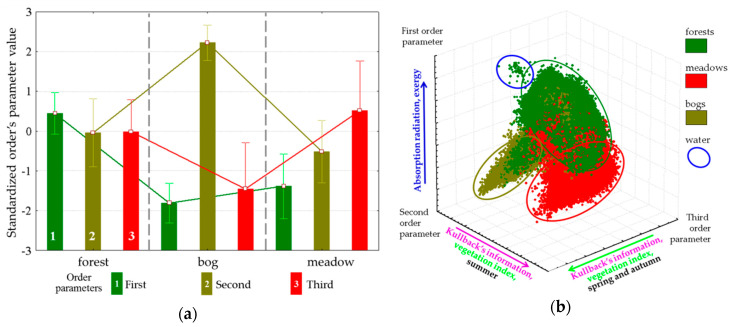
Order parameters for tree main ecosystem types: (**a**) Values of order parameters for types; (**b**) Types in space of order parameters.

**Figure 8 entropy-22-01132-f008:**
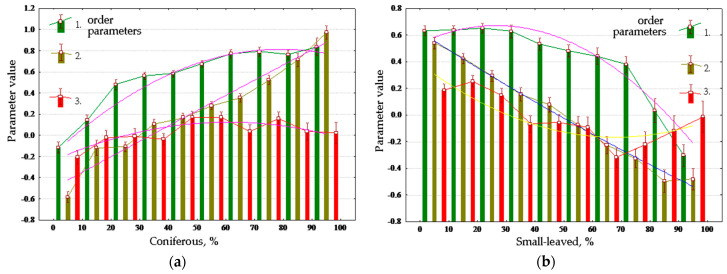
Vegetation properties and order parameters values: (**a**) Coniferous percent in stand; (**b**) Small-leaved percent in stand; (**c**) Herb coverage; (**d**) Moss coverage.

**Figure 9 entropy-22-01132-f009:**
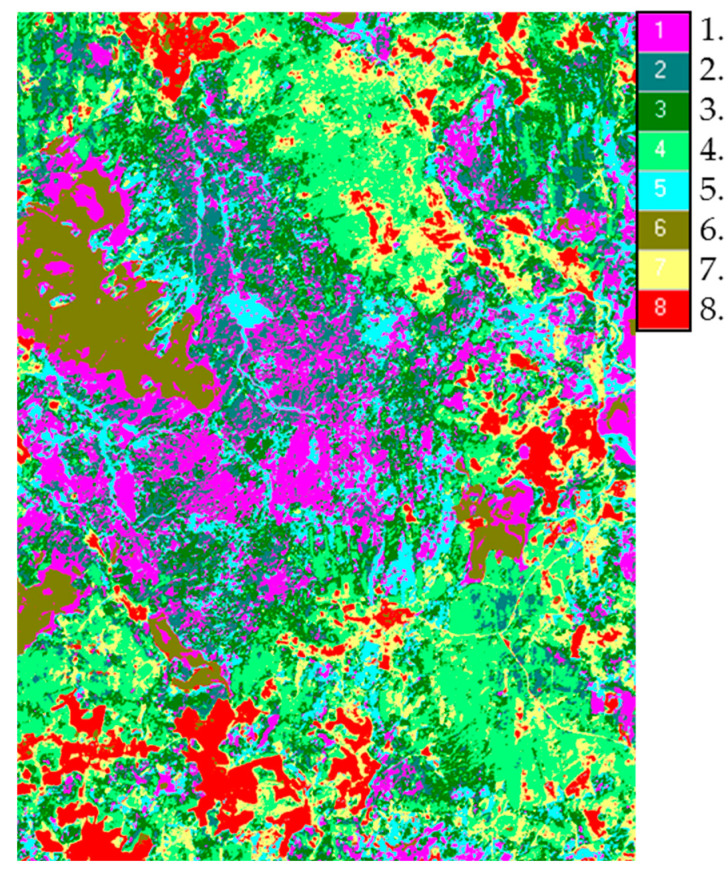
Classes of thermodynamic system based on ratio between order parameters: 1, old coniferous forest; 2, small-leaved and spruce forests; 3, mixed forests; 4, spruce-deciduous forests; 5, destroyed forests; 6, bogs; 7, meadows overgrown by shrubs; 8, hayfields, pastures.

**Table 1 entropy-22-01132-t001:** Used Landsat scene parameters.

Date				Scene’s Parameters		
Month	Day	Year	DOY ^1^	Sensor	Local Time	Sun Elevation, °
February	4	1987	35	Landsat 5 TM	11:08	13.35
	11	2007	42		11:42	18.18
March	21	1986	80		11:13	30.67
	22	2001	81	Landsat 7 ЕТМ +	11:38	32.13
	28	2003	87		11:36	34.22
April	10	2002	100		11:36	39.37
	21	2009	111	Landsat 5 TM	11:36	43.99
	27	2000	118	Landsat 7 ЕТМ +	11:45	45.93
May	3	1990	123	Landsat 5 TM	10:28	42.62
June	1	1992	152	Landsat 4 TM	10:54	49.85
	2	1995	153	Landsat 5 TM	10:54	49.97
	3	2007	154		11:42	54.06
	6	1988	158	Landsat 4 TM	11:15	52.18
	20	2002	171	Landsat 7 ЕТМ +	11:41	54.67
	21		172	Landsat 5 TM	11:23	53.37
August	22	2007	234		11:40	43.38
September	20	2006	263		11:41	33.35
	25	2008	269		11:32	30.44
	27	2000	271	Landsat 7 TM	11:36	30.43
October	15	1986	288	Landsat 5 TM	11:07	22.50

^1^ Day of year.

**Table 2 entropy-22-01132-t002:** Statistical parameters of thermodynamic variables for main community types: the top line represents the average, the bottom one shows the coefficient of variation (in bold type shown maximal meanings by ecosystem type for period).

Thermodynamic Variable	Vegetation Period			Summer		
	Forest	Bog	Meadow	Forest	Bog	Meadow
Absorbed radiation, W/m^2^	**106.3**	103.7	102.9	**115.8**	113.2	111.9
	17.0	17.2	**16.9**	6.4	**5.9**	6.6
Exergy, W/m^2^	**79.6**	74.3	73.2	**86.3**	80.8	79.0
	16.9	17.0	**16.9**	8.3	**6.8**	9.1
Temperature, °C	12.5	**15.9**	14.1	13.9	**17.9**	15.7
	46.1	40.5	**36.8**	41.9	32.7	**30.5**
Bound energy, W/m^2^	11.0	**11.7**	11.4	11.2	**12.2**	11.6
	7.6	10.5	**7.0**	7.8	8.3	**6.9**
Internal energy increment, W/m^2^	15.7	17.8	**18.3**	18.3	20.3	**21.2**
	34.5	**28.8**	30.5	17.9	14.9	**12.4**
Entropy of outcoming solar radiation, nit	1.46	1.48	**1.49**	1.45	**1.49**	1.48
	4.13	**3.01**	6.38	4.64	**2.06**	7.17
Information increment, nit	0.12	0.12	**0.16**	0.13	0.11	**0.15**
	46.88	**30.58**	53.68	46.55	**24.48**	53.69
Vegetation index, W/m^2^	4.41	5.16	**5.62**	5.09	5.58	**6.44**
	43.52	**25.12**	44.96	33.95	**18.92**	37.26

**Table 3 entropy-22-01132-t003:** Spearman’s correlations for order parameters and vegetation cover properties.

Vegetation Cover Properties		Order Parameters		
		1	2	3
Crown density by level	total density	0.205	−0.238	−0.348
	first level (35–25 m)	0.315	−0.115	−0.251
	second level (25–15 m)	0.251	−0.152	−0.295
	third level (15–5 m)	0.174	−0.084	−0.134
Tree height by level	mean height	0.366	−0.023	−0.085
	first level (35–25 m)	0.350	−0.025	−0.081
	second level (25–15 m)	0.358	−0.010	−0.149
	third level (15–5 m)	0.284	−0.04	−0.162
Basal stand area		0.356	−0.034	−0.265
Growing stock		0.386	−0.04	−0.237
Percent by tree species in total stand	coniferous	0.486	0.559	0.130
	small-leaved	−0.289	−0.480	−0.223
	broadleaf	−0.043	−0.240	0.025
Projective cover degree	understory	0.174	−0.002	0.051
	herb layer	−0.107	−0.177	−0.079
	moss layer	0.191	0.425	0.045

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
