# Peer review of "Solar Energy Transformation Strategies by Ecosystems of the Boreal Zone (Thermodynamic Analysis Based on Remote Sensing Data)"

_entropy, 2020, doi:10.3390/e22101132_

Round 1

Reviewer 1 Report

Dear colleagues,

congratulations for this excellent piece of work: it is a major advance in an important field. I read it through a few times, I still have to assimilate the large amount of data and results it contains, but I can understand that it is a remarkable article.

So, by all means, I suggest publication. The only things I note are

  1. The English is reasonably good as it is, but it can be improved.
  2. Check the heading of table 2 -- I think the term "Winter" is missing.
  3. At the end, when you cite "Gorshkov's biotic pump," good manners require that you cite also Gorshkov's coauthor, Anastassia Makarieva!
  4. I would suggest a more extensive discussion on the effects of climate change on the system being studied. As it is, it is confined to a single paragraph at the end -- as a reader I expected a little more after such a monumental effort!

And congratulations again!

Author Response

We deeply appreciate the kind review and appreciation of our study!

  1. Unfortunately, we are not native speakers, and the specificity of the text requires a highly qualified translation, which we did not dare to entrust to the translator. We have made some editing efforts.
  2. Table 2 gives exactly the growing season (without snow) and calendar (June, July, August) summer. Bringing the average values ​​of characteristics in winter for open surfaces - swamps and meadows) does not make sense, since they are completely covered with snow.
  3. We unconditionally accept this fair comment and correct accordingly.
  4. Thanks for the valuable recommendation! We tried to follow it. Also fulfilling the request of another reviewer, we supplemented the discussion with a comparison with other data add a Conclusion section.

Reviewer 2 Report

The article if really very interesting. There is an intersection of several theories, from the energy point of view , entropy optics and Exergy analysis basis. Nevertheless, the authors lost themselves in the introduction, because there is no connection and solid explanations.

For instance other articles published in entropy usually calls for minimum entropy production as a function of lifespan (Annamalai, Mady, Ozilgen).

I also think Professor Aoki applied the second law to an ecosystem in order to analyze similar perspectives, but without the same focus to use solar energy.

Other important point, reviewer knows that authors are one of the first to published something like this, nevertheless, there review from literature stoped before 2010 (one or two articles after). 

Therefore, for the article to be approved, it must contain some modifications in the literature review, also a better explanation from the theories of maximum energy use to minimal entropy production (Prigoggine).

In methods reviewer did not find any mistake, but, authors could use a nomenclature more often used in Exergy analysis 

Furthermore, Everstag was not the only one to carry out Exergy analysis of society, Dincer, Mosquim, and others also improved their concepts.

Author Response

We are deeply grateful to the reviewer for his deep, helpful and constructive feedback.

In the introduction we tried to say that there is no need to oppose Prigogine's principle of minimum entropy production and the principle of maximum entropy production. This was demonstrated in various ways by Leonid Martyushev (Doi: 10.1016 / j.physrep.2005.12.001) in his very famous work, and Levich (10.1142/9789812832092_0010) in a series of works on variational principles in biology and, in fact, in a series of works by Alexander Hazen. None of them refer to each other. The point is that we are talking about different interpretations of entropy and the relationship between them is a component of heat exchange / statistical entropy is a measure of disorder during heat exchange / entropy is a quantity of information that is transferred during communication processes, measure of particles' linkedness and interaction inside or around a system) .In fact, Jorgenson and Svirezhev in In their book (2004) tried to combine two views on this issue (phenomenological - Jorgenson, and Svirezhev - informational, through Kulbak - doi.org/10.1016/S0304-3800(01)00409) .Our calculation of the entropy of reflected solar radiation and on the spectrum gives us certainly not the classical phenomenological entropy. Hence our designations, which differ from the common exergy analysis. So, the entropy, which we estimate from the spectrum, is, in fact, the non-uniformity of energy absorption in different channels, therefore it is in no way thermodynamic, but informational. This leads to our preference for Hazen's works (unfortunately, his works of the 90s are practically not cited), who in his calculations, unlike Martyushev, operates more with an information interpretation of entropy.

The peculiarity of our approach is that we consider the informational interpretation of entropy to be more general than phenomenological. It is more invariant as information about a change in system states (trajectory selection) and a set of forbidden states (memory). At the same time, we did not find modern research outside our Russian-speaking team based on this approach to information / entropy. Strictly speaking, our analysis can be called not so much of an exergy as an informational / system analysis. We have tried to reflect these subtleties in the new edition of the introduction. We also tried to adhere to the designations from a series of our previous works based on remote-sensing approach.

We familiarized ourselves with the works on the estimation of the production of entropy by the human body (Özilgen, Prek, Mady, Annamalai) with great interest. These results generally confirm the results of Aleksey Zotin. As for ecosystems, at different ages and at different loads, the production of entropy is different - we observe different rates throughout the system life. At the same time, for ecosystems exchenging with the atmosphere, numerous studies show the maximization of the entropy flux from the ecosystem into the atmosphere (for example, the work of Alex Kleidon). All these studies operate with a phenomenological interpretation of entropy and their various results, lead us to a banal, in general, idea that the implementation of the maximum principle or the minimum principle is a function of the definition of the system and the method for calculating entropy.

In our literature review we focus on thermodynamical analysis of life matter and then skip works about ecological economics (Dincer, Mosquim) and a lot of another works about energy efficiency in economic “related” with ecology. Probably our understanding of ecology is to narrow, but we believe that firstly science about relation between species/communities and their environment.

Reviewer 3 Report

Entropy-934644

The main aim of the manuscript ‘Solar energy transformation strategies by ecosystems of the boreal zone (thermodynamic analysis based on remote sensing data)’ was not stated.

The manuscript has several typos and duplicated or missing words or punctuation marks. Besides, this manuscript has serious mistakes and weaknesses.

The Abstract section has to be improved. It is too short, and the essential information is missing.

The goal of the manuscript should be stated in the last paragraph of the Introduction section.

Figures and Tables should be explained by themselves. In Table 1, what does DOY mean?. Table 2, why are some numbers in bold? What does it mean?

Table 3 should be redesigned. To understand this Table is too hard.

The discussion section has to be improved. The first three paragraphs do not have citations. Discussión must explain the results alongside a comparison with the information of other authors. The Discussion section is not an extension of the Results section.

A Conclusion section is necessary.

Therefore, I recommend Major Revision.

Author Response

We are deeply grateful to the reviewer for the informative feedback.

  1. The reviewer points out that the main purpose of the article is not stated and rightly says that it should be placed in the last paragraph of the Introduction. We absolutely agree with this placement of the research objective. There it is located! Quote (page 4, line 147 - 150): “The foregoing allows us to determine the purpose of the presented study: as a test of the hypothesis of maximizing the flow of free energy at different time scales for specific ecosystems of the boreal zone of the European Plain ".
  2. We agree with the reviewer. We have expanded the section "Abstract".
  3. We have improved the indicated shortcomings of tables 1, 2 and 3.
  4. Of course, we understand the discussion section not only as an extension of the results section, however, in it we believe it is appropriate to begin with discussing and commenting on our results and drawing some conclusions based on them. This is our interpretation of the results and therefore not "Result", but at the same time it is too extensive to be in "Conclusion". It is rather difficult to compare our results with others because of the unique technique and incomplete coverage of the solar spectrum by the Landsat scanner measuring system. However, our estimates do not contradict the generally accepted ideas about the functioning of ecosystems and their interaction with the atmosphere. We tried to supplement the discussion with links to other sources and expand it. We have moved part of the discussion to a new section, Conclusion, created on your recommendation.

Round 2

Reviewer 1 Report

Revisions are OK -- thanks for the effort and congrats for this interesting study.

Reviewer 2 Report

In the first round of review, I did some appointments regarding not the quality of the work, but the organization. The article is very interesting and makes connections with several theories of thermodynamics and entropy. 

I think the new reader will follow better what the authors are trying to say. With the explanation given in the response. Moreover, the connection between several branches of second law analysis (with entropy) is more clear. 

The reviewer really liked this statement "All these studies operate with a phenomenological interpretation of entropy and their various results, lead us to a banal, in general, idea that the implementation of the maximum principle or the minimum principle is a function of the definition of the system and the method for calculating entropy." Authors could work in future analysis with this comprehensive analysis, which is not on literature yet.

Eventually, the reviewer does not agree with this statement: "The peculiarity of our approach is that we consider the informational interpretation of entropy to be more general than phenomenological." Nevertheless, this is science, and the authors made their point in the text and should be published in this discussion. This is at the center of the really healthy fight which this journal is carrying out. A comparison and formalization of the entropy by an information perspective and what is called more phenomenological. Nevertheless, this is the perspective of this reviewer and there is no mistake found in the article. 

Therefore congratulations to the authors, I really think this article will be a reference in future analyses of literature and deserves to be reproduced. I was really happy to review this article.

Reviewer 3 Report

The manuscript was improved. Currently, it could be accepted in the present form.